# Benefits of an Innovative 90-Day Longevity Workplace Program on Health in the United Arab Emirates (UAE)

**DOI:** 10.3390/ijerph22101594

**Published:** 2025-10-21

**Authors:** Ghanem Al Hassani, Erik Koornneef, Mariam Al Harbi, Salah El Din Hussein, Ghuwaya Al Neyadi, Omar Al Hammadi, Yasser Ghoneim, Mostafa Abdrabo, Stephen G. Holt

**Affiliations:** 1Abu Dhabi Health Services Company (SEHA) (A Pure Health Asset), Abu Dhabi P.O. Box 283572, United Arab Emiratessahussein@seha.ae (S.E.D.H.); sholt@seha.ae (S.G.H.); 2The Medical Office (TMO) (A Pure Health Asset), Abu Dhabi P.O. Box 283572, United Arab Emirates; 3College of Medicine and Health Sciences, Institute of Public Health, UAE University, Al Ain P.O. Box 15551, United Arab Emirates; 4College of Medicine and Health Sciences, Khalifa University, Abu Dhabi P.O. Box 127788, United Arab Emirates; 5Abu Dhabi Public Health Center, Abu Dhabi P.O. Box 5674, United Arab Emirates; 6Al Dhannah Hospital, Ruwais, Abu Dhabi, United Arab Emirates; 7Pharmaceutical Development Company (PDC), PDC-CRO, Dubai P.O. Box 500767, United Arab Emirates; 8SEHA Kidney Care, George Matthews Street, Al Mafraq, Abu Dhabi, United Arab Emirates

**Keywords:** health promotion, workplace wellness, health status, lifestyle, longevity

## Abstract

Unhealthy lifestyle behaviors, such as physical inactivity and an unhealthy diet, can decrease quality of life and increase the risk of obesity, depression, and chronic diseases like diabetes and cardiovascular disease. In workplace settings, these health issues are associated with increased healthcare costs and reduced productivity. The Pure Health 2K Longevity Study (PHLS) evaluated the effectiveness of a 90-day incentive-based lifestyle intervention among working adults in the United Arab Emirates (UAE). A single-arm interventional study was conducted by Abu Dhabi Health Services Company (SEHA) over a 4-month period. A total of 2300 participants aged 18–59 were enrolled, with 1688 (73.4%) completing the program. Participants underwent baseline and endline assessments, including physical measurements (weight, body mass index (BMI), waist circumference), biochemical parameters (blood pressure, glucose, Glycosylated hemoglobin (HbA1c), lipid profile, C-reactive protein (CRP), Gamma-glutamyl transferase (GGT), and self-reported health behaviors and adverse events. Significant reductions were observed in weight (77.0 to 75.9 kg), BMI (26.8 to 26.4 kg/m^2^), and waist circumference (95 to 93 cm) (all *p* < 0.001). Notably, 4.6% of participants transitioned from overweight to normal weight, and 3.4% from obese to overweight. No adverse events were reported. A short-term, workplace-based lifestyle intervention can produce meaningful improvements in anthropometric and biochemical health indicators, particularly among high-risk individuals.

## 1. Introduction

The World Health Organization suggests that 60% of the determinants of an individual’s health and quality of life are correlated with their lifestyle [1]. Unhealthy habits such as inactivity, poor diet, smoking, alcohol consumption, and stress have a significant impact on physical and mental health [2,3]. Lifestyle includes day-to-day behaviors and includes how people function during and outside their employment, but importantly includes exercise and diet [2]. Unhealthy eating habits and a sedentary lifestyle pose significant health risks worldwide [4]. They primarily contribute to noncommunicable diseases (NCDs), or “lifestyle diseases,” and global mortality [4,5]. Conversely, engaging in regular physical activity reduces the risk of cancer by 8–28%, cardiovascular disease and stroke by 19%, diabetes by 17%, and depression and dementia by 28–32%. It is estimated that 4–5 million deaths annually could be prevented if the global population was more physically active [5]. Furthermore, adopting a healthy diet and increasing exercise may improve long-term health by decreasing the likelihood of excessive weight gain or obesity, with a reduction in the risks of developing NCDs [5].

Chronic disease imposes a significant burden on individuals and results in diminished quality of life, early mortality, disability, and increased healthcare expenses [6]. Additionally, while chronic diseases were previously associated with older age groups, there is now a trend toward earlier onset, contributing to the economic burden due to reduced productivity from being absent from work (absenteeism) and decreased performance while at work (presenteeism) [3]. A systematic literature review revealed significant data on the employer’s burden of lost productivity for priority conditions [7].

The most common conditions leading to loss of productivity are pain (24%), cancer (22%), chronic lung disease (17%), cardiometabolic disease (16%), and depression (16%). Annual incremental costs of lost work productivity range between USD 100 to USD 10,000 per person, with higher costs associated with cancer, pain, and depression [7].

Workplace health promotion programs are emerging as a potential strategy in improving employees’ health and wellbeing, considering that adults spend a considerable amount of time of their lives in a workplace environment. A recent review looking at workplace-based health and wellbeing interventions [8] revealed a diverse range of approaches, including interventions focused on mental health and stress (36% of all interventions reviewed), followed by weight management and cardiometabolic health (25%). Effectiveness varied significantly between intervention types. Another recent systematic review reviewed workplace nutrition and health interventions and found positive results, including nutrition knowledge, enhanced self-efficacy, reduced risky behaviors, improved body mass index, and better blood biomarkers [9].

Health promotion and wellness programs can promote beneficial changes in employee health, potentially resulting in reduced healthcare expenses and enhanced employee wellbeing and productivity through a reduction in serious health conditions. A meta-analysis of studies related to workplace health programs showed that healthcare expenses could be reduced by USD 3.27 and absenteeism costs by USD 2.73 for every USD 1 invested in wellness programs [10]. Building on findings from a previous feasibility study [11], this larger follow-up study aimed to assess whether a short-term, incentive-driven workplace intervention could produce improvements in health indicators among a diverse working population.

## 2. Materials and Methods

### 2.1. Study Design and Participants

This single-arm interventional study was carried out to evaluate the health and wellbeing status of workers participating in an innovative 90-day incentive-based workplace health program and was a follow-up study to the pilot feasibility study [11]. It aimed to explore the impact of this program intervention on a broad group of employees aged 18 to 59 in the UAE. The study promoted a healthy lifestyle by integrating various health activities into daily routines designed to support employees’ overall health and wellbeing, encouraging them to make healthy choices and adopt positive lifestyle changes.

The study recruited any worker able to exercise and able and willing to provide informed consent. Exclusion criteria included advice from a healthcare practitioner not to exercise (regardless of type or intensity), having existing medical conditions that would prevent exercise (irrespective of type or intensity), pregnancy or breastfeeding, severe illness, severe joint or back injuries, planned major surgical procedures or other significant treatments during the project period, uncertainty or inability to fully commit to all the procedures and activities relevant to the project, current participation in another health promotion program, or completion of a health promotion program within the last 6 months.

The study received ethical approval from Abu Dhabi Department of Health and the UAE Ministry of Health and Prevention Research Ethics Committee with reference number DOH/CVDC/2023/913 on 21 May 2023. Informed consent was obtained from all participants before starting.

The study was conducted with participants from across several organizations in the UAE: Pure Health Group, one of the largest healthcare networks in the region, including Abu Dhabi Health Services Company (SEHA), as well as a large energy company headquartered in the UAE. Participation was based on employee self-enrollment following internal promotion of the program via internal communication channels by each participating organization, including registration links sent to all.

Before and after the intervention, all participants underwent a comprehensive baseline and follow-up assessment including demographic information, medical history, anthropometry, and some blood tests. Anthropometric and biochemical data were collected at baseline (day 1) and follow-up (day 90) at on-site health stations. Weight, height, BMI, waist circumference, and blood pressure (systolic and diastolic) were measured by trained healthcare staff using standardized equipment and protocols. BMI categories were defined using WHO standards: underweight (UW, below 18.5 kg/m^2^), normal (NW, 18.5–24.9 kg/m^2^), overweight (OW, 25.0–29.9 kg/m^2^), and obese (OB, ≥30.0 kg/m^2^).

Blood samples were collected by licensed clinical staff and samples were analyzed in accredited, licensed laboratories. Participants also completed a survey at both time points, which included self-reported physical activity (steps per day, exercise frequency) and wellbeing (sleep quality, mood).

The participants then followed the 90-day Pure Health Longevity Program (PHLP) including activities and challenges related to physical, mental, general health, nutrition, and health education, as well as social challenges. PHLP encompassed a range of interventions, including healthy eating/healthy living promotion, physical activity tracking (e.g., step count), and stress management. These components were monitored through a self-administered mobile application (developed by Pure Health Group, called PURA, V0.7.1) and wearable devices (Fitbit Charge, Pure Health Medical Supplies, Dubai, United Arab Emirates).

Once registered on the PURA app, the participants were able to participate in daily challenges, in terms of calories burnt and physical activity (step count). For each challenge that was successfully completed, the participants earned so-called Fitcoins, which could be converted into vouchers at the end of the program, depending on the participant’s level of engagement. In addition, the participants were able to view and monitor their own and their peers’ performance on a leadership board on the mobile application (each participant competed within their own functional team; in total, there were 33 different functional units).

At the end of the intervention period, participants underwent a similar follow-up assessment to that at the baseline. Patients were classified at baseline (pre) and at the end of the study (post) based on their BMI into obese (OB), overweight (OW) normal weight (NW), and underweight (UW).

### 2.2. Study Outcomes

The primary endpoint of the study was to show a reduction in BMI by an increase in physical activity during the study. Secondary outcomes included changes in other anthropometric measurements (weight, body mass index (BMI), waist circumference), changes in self-reported health questionnaires, and changes in blood tests associated with adverse outcomes.

### 2.3. Statistical Analysis

Data were analyzed using GraphPad Prism V10.4.2. Normality was assessed using D’Agostino and Pearson tests. As most variables were not normally distributed, non-parametric tests were used (Kruskal–Wallis for unpaired group comparisons and Wilcoxon for paired data). Statistical significance was assigned if *p* < 0.005, assuming two-sided significance testing.

## 3. Results

### Baseline Characteristics of Study Participants

A total of 2300 participants were enrolled in the study, all receiving intervention, without a control group.

A total of 1700 participants completed the 90-day program and were followed up for efficacy data collection after 90 days of enrolment in the program. In total, 1688 participants were included in the final analysis, as data from 12 records were excluded due to incomplete anthropomorphic data and these were included in the analysis of the dropout group (DO) (Appendix A, Figure A1 and Figure A2). Dropouts were more likely to be younger, female, smokers, and Emirati and have a higher BMI. Baseline characteristics of the full cohort (FC), those who completed (C) the study, and DO are seen in Table 1. There was a high proportion of participants from the Asian subcontinent (Appendix A, Table A1).

Compared to the baseline, there was a significant difference in all anthropometric measures including weight, BMI, and waist circumference (Figure 1).

For the primary endpoint, there was a significant reduction in weight (Figure 1A) from pre-intervention (median [interquartile range]) at 77.0 [68.0–87.0] kg to post-intervention at 75.9 [67.2–85.7] kg (*p* < 0.0001). BMI also significantly reduced from 26.8 [24.4–29.8] to 26.4 [24.1–29.4], *p* < 0.0001 (Figure 1B). Waist measurements significantly reduced from 95 [88.5–103] to 93 [86–101] cm, *p* < 0.0001 (Figure 1C). Weight loss was mainly seen in those who were OB or OW at baseline (Figure 1D). There was no significant influence of gender on these outcomes (Appendix A, Figure A2). Whilst males were significantly heavier than females pre- and post-intervention, reduction separated by gender failed to show statistical significance (Appendix A, Figure A2a). Similarly, BMI failed to show significance (Appendix A, Figure A2b). Statistically significant changes in waist circumference, however, were seen in men and women (Appendix A, Figure A2c).

Further sub-analysis looked at these groups to classify benefit; overall, there was a reduction in OB and OW patients and an increase in NW subjects (Appendix A, Figure A2a).

The number of OB subjects pre-intervention was 401 and this fell to 364 by the end of the study; similarly, the OW patients numbered 751 at the start of the study but had reduced to 734 by the end, with the number of patients classified as NW increasing from 528 to 581. The median BMI in the OB, OW, and NW groups reduced significantly during the study (Figure 2a). Although overall most patients remained in the OW category, 78 (4.6%) patients moved from OW to NW and 57 (3.4%) moved from OB to OW. There were a few other subjects who changed BMI class (Table 2).

Although only some subjects responded to questions on self-reported exercise pre and post, 24.1% of participants self-reported an increase in moderate activity minutes and 27.2% in strenuous activity minutes following the intervention.

OB and OW subjects lost far more weight than those who were NW or UW (Figure 1D), with the former groups losing a median of 0.85 [−2.3–0.5] kg compared with the latter whose median weight loss was −0.15 [−1.2–0.7] kg, *p* < 0.0001. Patients starting in the OB, OW, and NW categories all had statistically significant changes in their BMI at 90 days (Figure 1C). We further examined changes in the waist circumference of the different groups at baseline and follow-up compared with those who were classified as NW at baseline and follow-up (Figure 2b). Those who went from obese to overweight (OB_OW) and those going from overweight to normal weight (OW_NW) had significant reductions in their waist circumference, whilst those who went from normal weight to overweight (NW_OW) increased their waist circumference.

There were correlations between weight, body mass index (BMI), and waist circumference (Appendix A, Figure A3) with systolic and diastolic blood pressures (SBP and DBP). The BMI groups are shown along with the median systolic and diastolic BPs (Figure 3a,b).

There was an overall significant correlation of BMI categories with systolic and diastolic blood pressure (SBP and DBP) (Figure 3) and overall on linear regression (Appendix A, Figure A3). 

In terms of blood biomarkers associated with adverse cardiometabolic risk, low-density-lipoprotein-associated cholesterol (LDL), triglycerides (TG), fasting blood glucose (FBG), and glycosylated hemoglobin (HBA1c) were significantly different in each BMI group (Figure 4).

Other biomarkers associated with adverse outcomes, and which correlated with BMI, included c-reactive protein (CRP) and gamma-glutamyl transferase (GGT) (Figure 5).

C-reactive protein (CRP) and gamma-glutamyl transferase (GGT) across BMI groups. Both biomarkers showed significant correlation with BMI levels.

Additionally, self-reported step count data was available for 1510 participants. Of these, 50% (*n* = 752) reported an increase in daily steps over the 90-day period, while 42% maintained similar levels, and only 8% reported fewer steps. This aligns with broader trends in increased physical activity reported elsewhere in the study and suggests participants became more active overall, which may have contributed to improvements in anthropometric and biochemical outcomes.

Finally, no adverse events or serious adverse events were reported throughout the study period.

## 4. Discussion

The 2K Longevity Trial promoted healthy lifestyle choices through various health activities; the holistic approach aims to support employees’ overall health and wellbeing. The main aim of the study was to target a modest weight reduction by an incentive-driven and sustainable exercise program. Weight loss in overweight subjects [12] and exercise [13] have independent health benefits. The health risks of being OB or OW are well documented [14].

More subjects who responded to the questionnaires reported increasing their moderate and strenuous activity but only a minority had data in this domain. Additionally, there was a marked increase in self-reported daily step counts, which appears to indicate that the program was successful in those who participated and likely this along with health advice contributed to the achievement of the primary endpoint. The most significant effects were in the obese (OB) and overweight (OW) groups, compared with those already at normal weight (NW), although we would argue that benefits in those already at normal weight are likely to be harder to achieve, but nonetheless may be valuable.

Anthropometric indicators proved an important discriminator of many biochemical tests which have implications for long-term health. This study showed that BMI was significantly associated with higher BP, LDL, FBG, HBA1c, CRP, and GGT.

The American Diabetes Association has stated that exercise and weight loss can lower blood glucose by increasing insulin sensitivity and helping muscles to use glucose for energy, with or without insulin, and can also lower HbA1c over time [15]. Research indicates that maintaining optimal blood glucose levels can lead to a longer life. In contrast, chronic high blood sugar can accelerate aging and make individuals more susceptible to age-related diseases [16,17].

Effectively managing weight, blood sugar, blood pressure, and cholesterol can potentially extend the life expectancy of the average person with type 2 diabetes by 3 years. For individuals with the highest levels of BMI, HbA1c, LDL, and systolic blood pressure, lowering these levels could potentially result in a life expectancy increase of over 10 years. Additionally, reduced BMI on average was associated with the greatest gain in life expectancy, followed by reduced HbA1c [18]. This program is thus likely to be successful if these results are sustained. Indeed, we demonstrate clearly here that the higher-BMI group is associated with marked differences in blood pressure, lipids, glucose handling, inflammation, and GGT (a potential marker for non-alcoholic steatohepatitis). Clinicians can illustrate the value of attaining a normal BMI to motivate patients. It also suggests that such wellness programs should be prioritized for those in the higher-BMI groups.

Other studies have identified biochemical markers associated with exceptional longevity, including lower levels of glucose, CRP, and GCT. One such cohort study found that reaching 100 years of age was associated with some biochemical markers including lower glucose, CRP, and GGT [19]. Encouragingly, our study showed that following the 90-day program, there was a significant decrease in LDL and TG levels. Moreover, several studies have recognized plasma TG as a potential biomarker for aging [20], and TG levels may be inversely associated with cognitive function in older adults without dementia [21].

In this study, hs-CRP levels were significantly correlated with BMI. Since systemic inflammation may serve as a primary risk factor for various diseases, such as neurodegenerative and cardiovascular diseases and tumors. Low levels of inflammation may play a significant role in predicting long-term survival and this includes CRP [22]. Studies have shown that the Japanese have lower proinflammatory protein levels than American and European populations, which may be linked with body habitus, cultural, and lifestyle practices [23].

Raised GGT may indicate hepatic damage, including non-alcoholic steatohepatitis [24], and there was a significant association here of GGT with BMI at both baseline and follow-up. Given its emerging role as a potential biomarker of aging and metabolic risk, these findings support the notion that lifestyle-driven reductions in BMI could contribute to improved hepatic health as well as preventing other age-related diseases.

After intervention, many participants had reduced their BMI class (with only a few showing the inverse) indicating that this change, if sustained, is likely to be beneficial for their long-term health. It is perhaps disappointing that we showed relatively low statistical significance in biomarker changes for those subjects who reduced their BMI (Appendix A, Figure A4). The only significant result was that for HbA1c which reduced significantly in those going from OB to OW, compared with those who remained normal weight. However, the overall relatively low numbers in these groups are likely to mean that we lack the statistical power to show this in only 90 days. We are hopeful that if these individuals continue with this program, they will in time show such anthropomorphic and biomarker changes. However, we are cognizant that there is often a significant ongoing attrition rate in program participation [25].

No adverse events or serious adverse events were reported throughout the study period. However, it is of note that one patient moved from NW to UW. Since only a few patients were UW, we did not examine this group in detail, but we may need to consider whether such persons should be excluded from similar programs, as there are well-described health risks for being underweight [26].

This study has several limitations. The most significant is the high dropout rate and thus indication bias—with dropouts more likely to be female, Emirati, younger and heavier (Appendix A, Figure A5). This suggests that the program and its interventions may not have met the expectations and needs of these subgroups. These dropouts arguably have more to gain from programs like this; therefore, it is important to design and implement tailored programs that focus on the needs of this group to engage in workplace programs. Missing data and dropouts may bias results; dropouts were more likely to be younger, female, smokers, and Emirati, which may limit generalizability.

Additionally, missing data in certain domains may have reduced the statistical power and increased the risk of possible type I/II errors. The reliance on self-reported data, particularly related to physical activity, is difficult to verify and we did not try to further analyze the subgroups with this data. Future studies will try to ensure that participants respond to as many questions as possible and we hope to conclusively quantify the actual changes in exercise using subject wearables.

## 5. Conclusions

This 90-day longevity program resulted in significant improvements in anthropometric measures, with a significant reduction in obesity and increases in healthy weight, especially in the OB and OW groups who appeared to benefit the most. Biomarker profiles were associated with BMI and became more advantageous towards those in the NW group. Importantly, the program appeared safe without reported adverse events, but we may need to consider a different program for those who are UW at baseline. We had a significant dropout rate, and we need to work hard at targeting such individuals as they may have more to gain from such programs.

Targeted wellness programs aimed at groups may be highly effective in promoting healthier lifestyles and may lead to better employee health, which ultimately may significantly reduce healthcare costs to both the employer and the payer.

## Figures and Tables

**Figure 1 ijerph-22-01594-f001:**
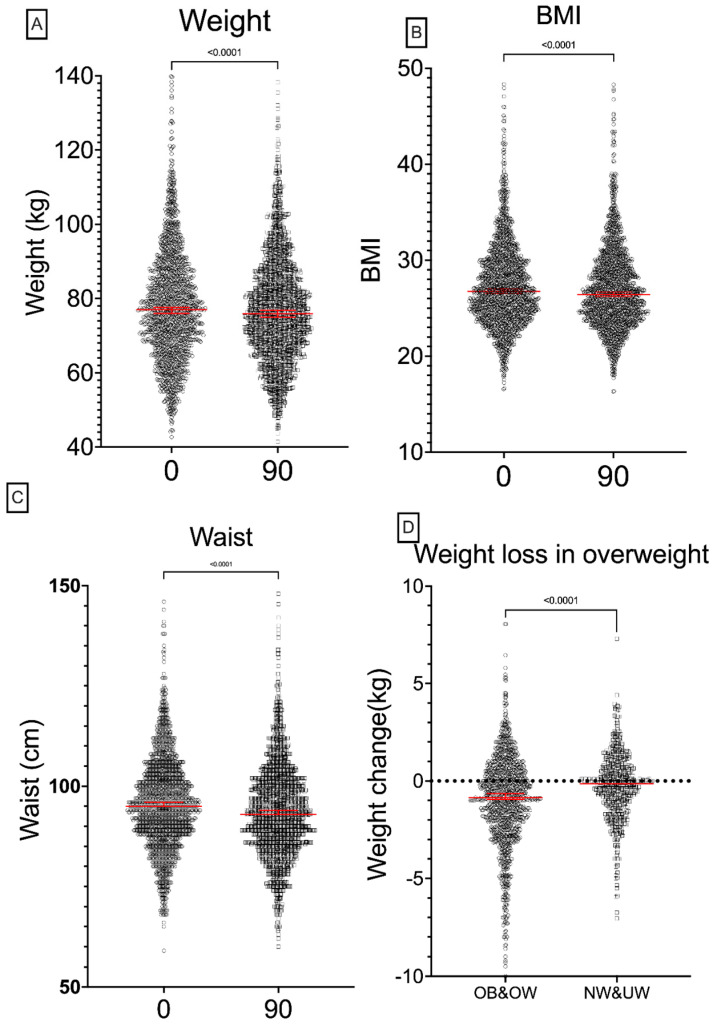
Changes in weight, BMI, and waist circumference after the 90-day program. Statistically significant reduction in overall weight (**A**), BMI (**B**), and waist circumference (**C**) after the 90-day program. Panel (**D**) shows weight loss was primarily observed in the overweight (OW) and obese (OB) groups, dotted line indicates no change. Wilcoxon paired tests were applied in (**A**–**C**), and Mann–Whitney tests in (**D**) with unpaired columns. Median values and 95% confidence intervals are shown in red.

**Figure 2 ijerph-22-01594-f002:**
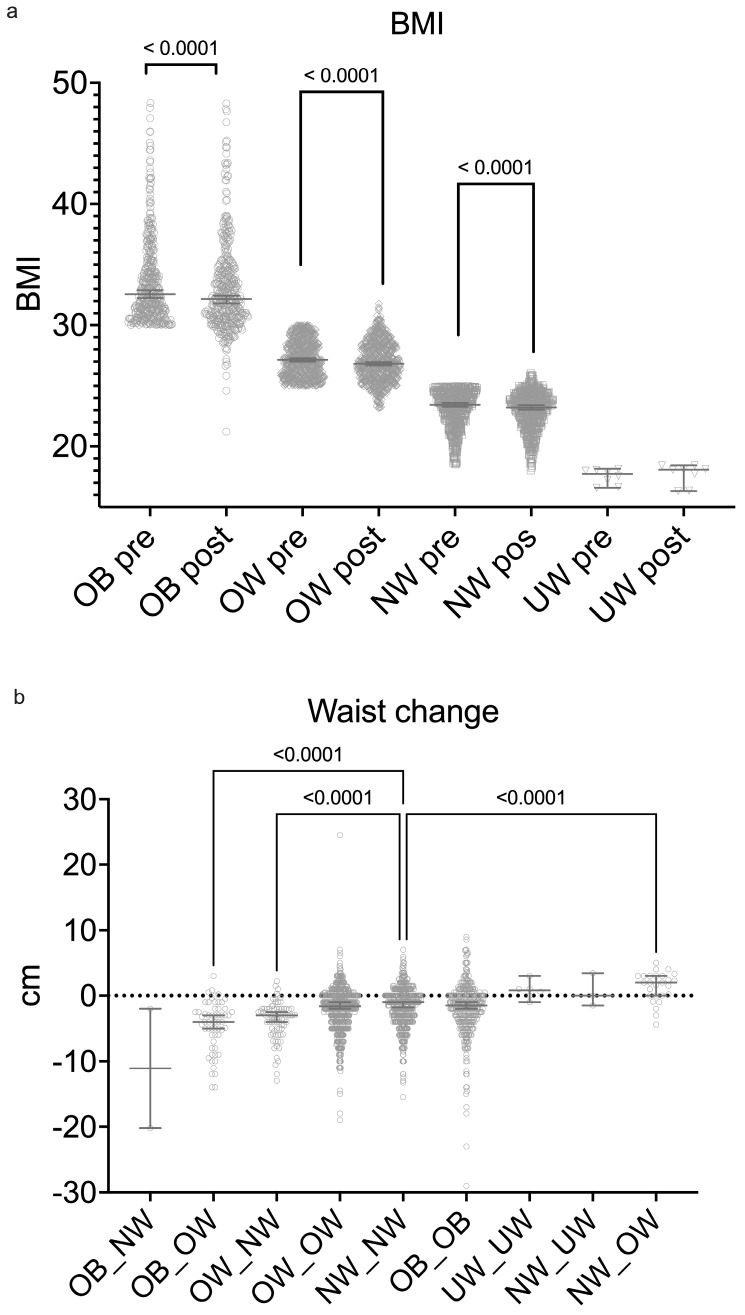
BMI and waist circumference changes by weight category before and after the 90-day program. (**a**) Change in BMI across groups pre- and post-intervention in each BMI group. Significant reductions in all groups except UW. (**b**) Waist circumference changes across groups. Significant reductions occurred in the OW and OB groups, while the NW and UW groups showed minimal change. Median values and 95% CIs shown. Dotted line shows no change in waist circumference.

**Figure 3 ijerph-22-01594-f003:**
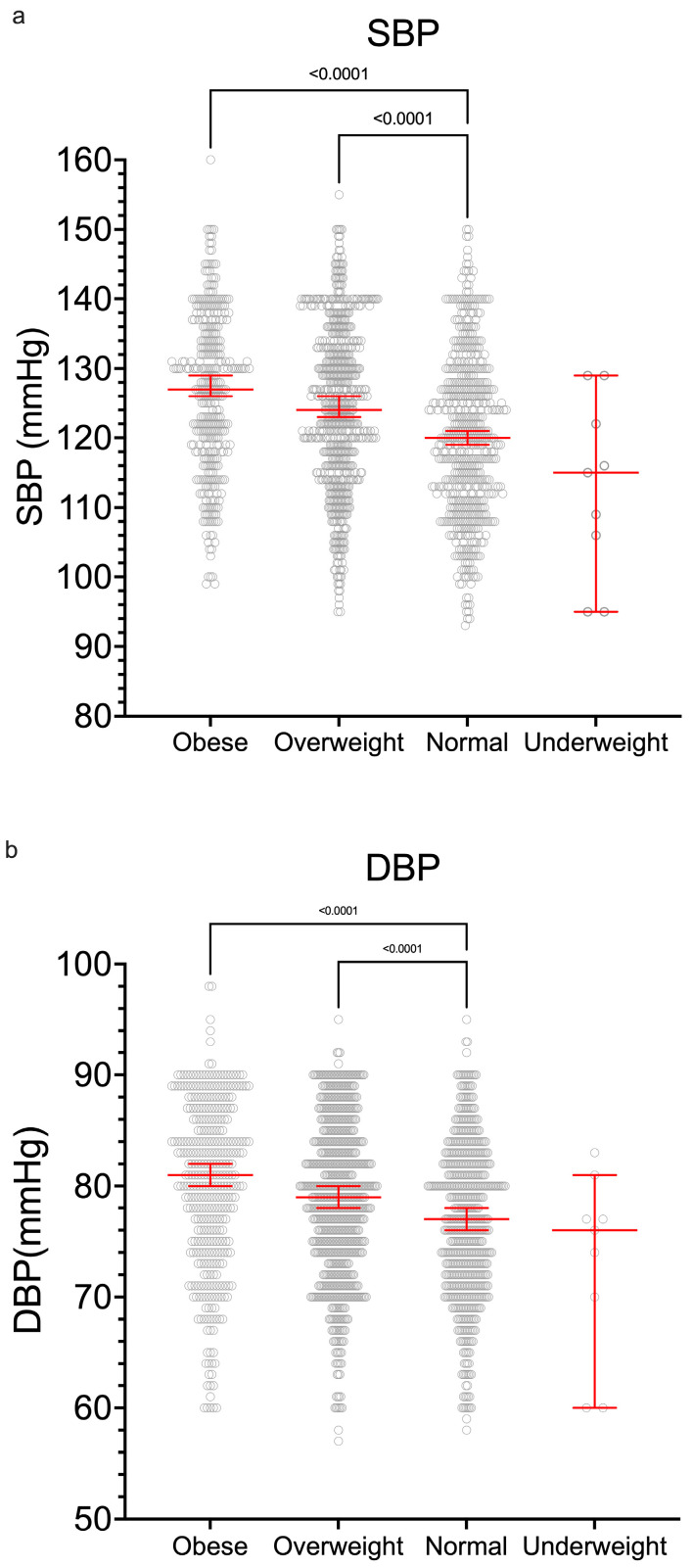
There was a correlation of BMI categories with (**a**) systolic and (**b**) diastolic blood pressure. Median values and 95% CIs are shown; Kruskal–Wallis tests were applied.

**Figure 4 ijerph-22-01594-f004:**
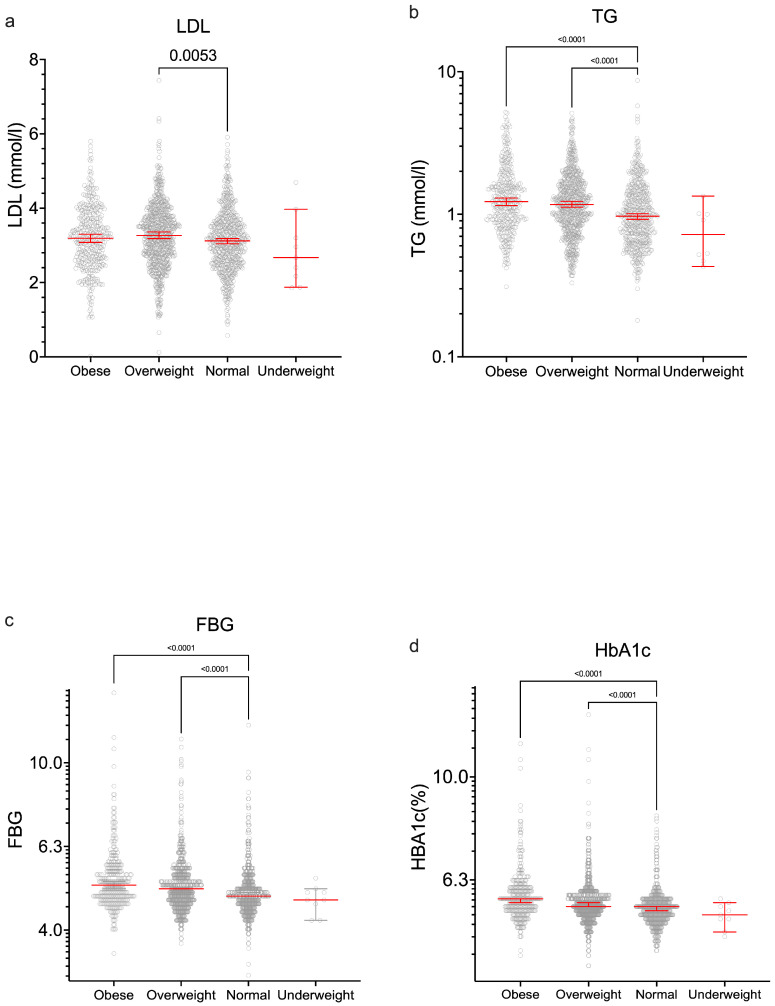
Baseline biomarkers (**a**) LDL, (**b**) triglycerides, (**c**) fasting glucose, and (**d**) HbA1 across BMI categories. Median values and 95% CIs shown. *p*-values from Kruskal–Wallis tests. Graphs showing health markers at pre-intervention (day 0) in the different BMI groups: obese (OB), overweight (OW), normal weight (NW), and underweight (UW).

**Figure 5 ijerph-22-01594-f005:**
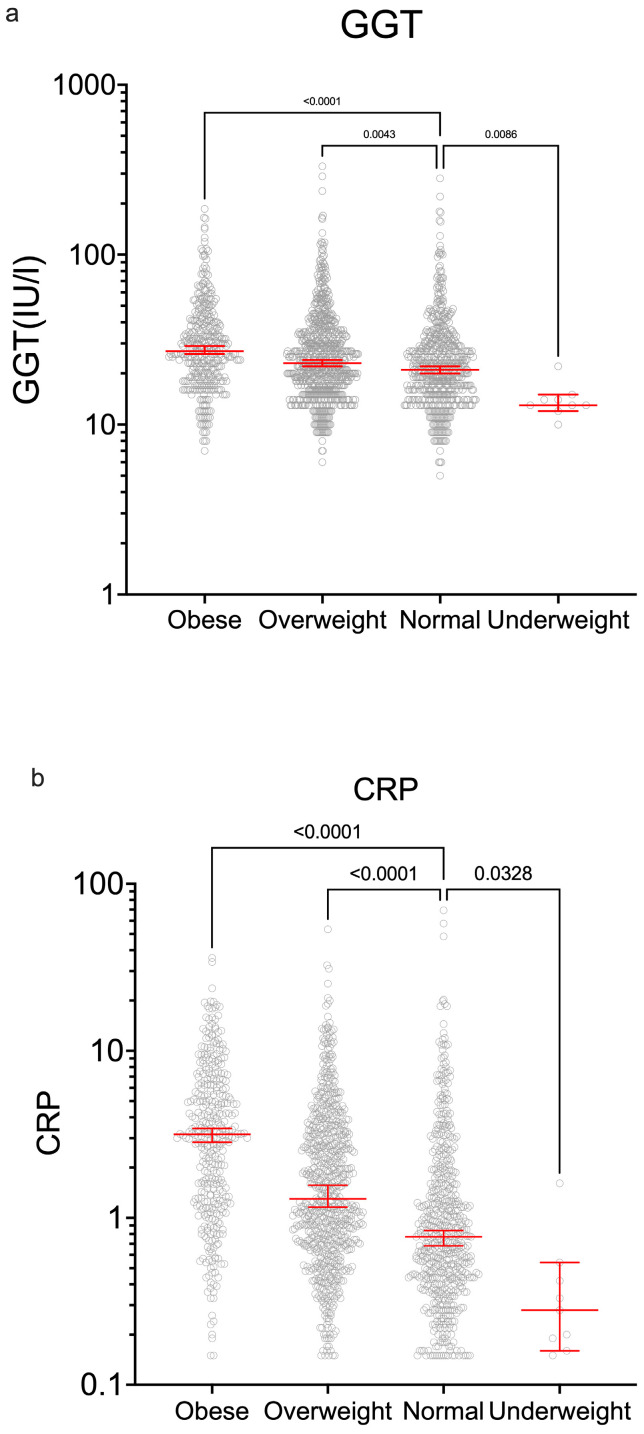
There were significant differences in liver enzymes and inflammation across groups (**a**) gamma-glutamyl transferase (GGT) and (**b**) C-reactive protein (CRP) levels across BMI categories.

**Table 1 ijerph-22-01594-t001:** Baseline characteristics of study participants: median [interquartile range].

	Full Cohort	Complete	Dropouts	*p* Complete vs. Dropouts
*N*	2300	1688	612	
Age (Y) (MED [IQR])	39 [33–45]	39 [34–45]	37 [32–43]	<0.0001
Weight (kg) (MED [IQR])	77.5 [68.0–88]	77.0 [68.0–87.2]	79.1 [68.3–91.1]	<0.0001
Male/Female Ratio	2:1	2.2:1	1.7:1	<0.0001
Height (cm) (MED [IQR])	169 [162–175]	169 [162–175]	169 [162–175]	not significant
BMI (kg/m^2^)	27.0 [24.5–30.1]	26.8 [24.4–29.9]	27.2 [25.0–30.8]	<0.001
UAE National/Emirati (%)	17.4	9.7	46	
Non-Smoker/Smoker Ratio	4.4:1	5.2:1	3.1:1	

**Table 2 ijerph-22-01594-t002:** Transitions in BMI categories from baseline to post-intervention.

Group	Abbr.	*n*	%
Total		1688	
OVERWEIGHT-OVERWEIGHT	OW_OW	651	38.6
NORMALWEIGHT-NORMALWEIGHT	NW_NW	499	29.6
OBESE-OBESE	OB_OB	342	20.3
OVERWEIGHT-NORMALWEIGHT	OW_NW	78	4.6
OBESE-OVERWEIGHT	OB_OW	57	3.4
NORMALWEIGHT-OVERWEIGHT	NW_OW	26	1.5
OVERWEIGHT-OBESE	OW_OB	22	1.3
UNDERWEIGHT-UNDERWEIGHT	UW_UW	6	0.4
NORMALWEIGHT-UNDERWEIGHT	NW_UW	3	0.2
OBESE-NORMALWEIGHT	OB_NW	2	0.1
UNDERWEIGHT-NORMALWEIGHT	UW_NW	2	0.1

## Data Availability

The data presented in this study are available on request from the corresponding author due to privacy and legal restrictions.

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
