# Peer review of "Benefits of an Innovative 90-Day Longevity Workplace Program on Health in the United Arab Emirates (UAE)"

_ijerph, 2025, doi:10.3390/ijerph22101594_

Round 1
Reviewer 1 Report
Comments and Suggestions for Authors
Thank you for the opportunity to review your manuscript ‘Benefits of an Innovative 90-Day Longevity Workplace Program on Health in the United Arab Emirates (UAE)’. I enjoyed reading this paper and find the work of interest to the readership of the journal. The applied intervention presented in the paper appears to be a follow up study to the pilot feasibility study by the authors (reference number 8). Whilst the paper is interesting, the paper requires some major revisions. In particular, the authors need to revise the results section of the paper as it is currently difficult to read and incomplete. The authors should also work to strengthen the discussion section as it does not do justice to the paper and lacks wider considerations surrounding the field of behaviour change. The authors could also benefit from optimising their paragraph structure to improve the flow and readability of the paper, as well as discussing the diversity of their sample.
Specific feedback as follows:
Lines 44-45 require a supporting reference
Please clarify lines 55-56. Are you referring to a systematic literature review you have published or a published paper? Presumably the latter in which case provide the reference. If the former, please change this and cite published work.
I would like an additional sentence at end of introduction section (line 66) as this section ends very abruptly and does not lead into subsequently section clearly enough. Please add in a sentence which draws together justification for the health programme you have done.
Regarding methods and measures, I read the authors pilot study paper but still some information was lacking on measures. Please be clearer and transparent.
Overall, the results section is interesting but needs some major revisions to be clearer to the reader.
Table 1 could be presented more clearly for the reader.
Regarding demographics, it appears from Table 1 that only a small proportion of your sample are UAE national/Emirati. Where are the rest of the sample from? Please share this data. And why such a high drop out from this group?
I’m also unclear why male: female ratio was provided? Please just include n and %. To this extent, I am inferring there are 2x as many males in the sample, so did you look at gender in your analyses? This is lacking – please address.
For Figure 1, the x axis is not labelled. I presume this is days (90). As a wider point, you can just refer to pre and post intervention which would probably be more helpful.
Figures 2 and 3 are unreadable. Please revise these to more effectively convey information. The authors make reference to Figures 2a-d but these are not clearly labelled.
Lines 188-190 is unhelpful and seems out of context. Either explore this data more fully and meaningfully or don’t. It felt like this was copied and pasted from elsewhere.
The discussion reads as disjointed and needs strengthening. Whilst it is important to consider cross cultural differences, I’m confused as to why the authors single out Japan (lines 234-237). You need to make a wider point here about the limitations of YOUR data sample and the implications, as well as wider cross cultural differences.
Why is line 228-229 a standalone sentence? Paragraphs please. Likewise, lines 242-244.
Please clarify by what you mean in lines 263-265 ‘There was also a lack of data for some domains which may have led to possible type I/II errors and future studies will try to ensure that participants respond to as many questions as possible. Self-reported data is difficult to verify, so we did not try to further analyse the subgroups with this data.’
I would like to authors to further acknowledge the limitations of a 90 day behaviour change intervention. Please link to literature regarding sustained behaviour change and discuss time frames.
I’m not sure abbreviations list is needed (unless this is a requirement of the journal).
The information in Figures A2 & A3 could be more effectively presented (perhaps in different format) for the reader. Currently, it is hard to meaningfully interpret these figures.
Author Response
|
Reviewer 1 |
Comments |
Authors’ Comments |
|
1 |
Thank you for the opportunity to review your manuscript ‘Benefits of an Innovative 90-Day Longevity Workplace Program on Health in the United Arab Emirates (UAE)’. I enjoyed reading this paper and find the work of interest to the readership of the journal. The applied intervention presented in the paper appears to be a follow up study to the pilot feasibility study by the authors (reference number 8). Whilst the paper is interesting, the paper requires some major revisions. In particular, the authors need to revise the results section of the paper as it is currently difficult to read and incomplete. The authors should also work to strengthen the discussion section as it does not do justice to the paper and lacks wider considerations surrounding the field of behaviour change. The authors could also benefit from optimising their paragraph structure to improve the flow and readability of the paper, as well as discussing the diversity of their sample. |
Thank you for the feedback, we have incorporated the feedback in the new, revised manuscript (v2). |
|
2 |
Lines 44-45 require a supporting reference. |
Done. |
|
3 |
Please clarify lines 55-56. Are you referring to a systematic literature review you have published or a published paper? Presumably the latter in which case provide the reference. If the former, please change this and cite published work. |
Citation is included in the new, revised manuscript (v2). |
|
4 |
I would like an additional sentence at end of introduction section (line 66) as this section ends very abruptly and does not lead into subsequently section clearly enough. Please add in a sentence which draws together justification for the health programme you have done. |
These comments has been incorporated in the new, revised draft manuscript (v2). |
|
5 |
Regarding methods and measures, I read the authors pilot study paper but still some information was lacking on measures. Please be clearer and transparent. |
Thank you for highlighting this, incorporated in the new, revised manuscript (v2). |
|
6 |
Overall, the results section is interesting but needs some major revisions to be clearer to the reader. |
Thanks for your comments. The results of the intervention are promising. We have further amended the manuscript, incorporating the reviewer’s comments. |
|
7 |
Table 1 could be presented more clearly for the reader. |
The improved formatting has addressed this to aid review. |
|
8 |
Regarding demographics, it appears from Table 1 that only a small proportion of your sample are UAE national/Emirati. Where are the rest of the sample from? Please share this data. And why such a high drop out from this group? |
We have added and additional table in the Appendix detailing nationalities. We have also added a paragraph why the Emirati dropout rate was higher than other groups. |
|
9 |
I’m also unclear why male: female ratio was provided? Please just include n and %. To this extent, I am inferring there are 2x as many males in the sample, so did you look at gender in your analyses? This is lacking – please address. |
There is a gender imbalance in the expatriate population in Abu Dhabi and this reflects the population demographic prevalent here. |
|
10 |
For Figure 1, the x axis is not labelled. I presume this is days (90). As a wider point, you can just refer to pre and post intervention which would probably be more helpful. |
Figure has been changed, as requested. |
|
11 |
Figures 2 and 3 are unreadable. Please revise these to more effectively convey information. The authors make reference to Figures 2a-d but these are not clearly labelled. |
Perhaps the pdf figures do not properly resolve the figures at low resolution, but the high-quality figures supplied are completely legible. We can separate the figures, but this may take up too much space. We could pick two or four graphics and move the rest to supplementary figures if required. |
|
12 |
Lines 188-190 is unhelpful and seems out of context. Either explore this data more fully and meaningfully or don’t. It felt like this was copied and pasted from elsewhere. |
This has been addressed in the new, revised manuscript (v2). |
|
13 |
The discussion reads as disjointed and needs strengthening. Whilst it is important to consider cross cultural differences, I’m confused as to why the authors single out Japan (lines 234-237). You need to make a wider point here about the limitations of YOUR data sample and the implications, as well as wider cross cultural differences. |
This has been addressed in the new, revised manuscript (v2). |
|
14 |
Why is line 228-229 a standalone sentence? Paragraphs please. Likewise, lines 242-244. |
This has been addressed in the new, revised manuscript (v2). |
|
15 |
Please clarify by what you mean in lines 263-265 ‘There was also a lack of data for some domains which may have led to possible type I/II errors and future studies will try to ensure that participants respond to as many questions as possible. Self-reported data is difficult to verify, so we did not try to further analyse the subgroups with this data.’ |
This has been addressed in the new, revised manuscript (v2). |
|
16 |
I would like to authors to further acknowledge the limitations of a 90 day behaviour change intervention. Please link to literature regarding sustained behaviour change and discuss time frames. |
See new, revised text in the Discussion section |
|
17 |
I’m not sure abbreviations list is needed (unless this is a requirement of the journal). |
Noted, we included the abbreviations to enhance the readability. |
|
18 |
The information in Figures A2 & A3 could be more effectively presented (perhaps in different format) for the reader. Currently, it is hard to meaningfully interpret these figures. |
Amended all figures to improve readability. We also removed some graphs and put in text |
Reviewer 2 Report
Comments and Suggestions for Authors
Review Report
Manuscript Title: Benefits of an Innovative 90-Day Longevity Workplace Program on Health in the United Arab Emirates (UAE)
Thank you for the opportunity to review this manuscript. The study investigates the effects of a 90-day workplace-based lifestyle intervention on various cardiometabolic outcomes. This study addresses a topic that is both timely and relevant. However, the manuscript has several critical limitations.
The manuscript lacks an adequate review of prior research and a contextual discussion that would help clarify the rationale and implications of the study. In addition, the Methods section is severely underdeveloped. The authors do not provide sufficient detail regarding how participants were recruited or what the intervention entailed, making it difficult for readers to understand how the study was conducted.
Introduction
-
While several previous studies have examined the effects of workplace-based lifestyle interventions on health, the current manuscript does not sufficiently reference or discuss these findings. The Introduction should be expanded to include a comprehensive overview of relevant prior research and to clearly highlight the research gap this study aims to address.
-
The aim or objective of the study is not clearly stated in the final paragraph of the Introduction
Methods
-
There is a typographical error: "This single-arm study observational study". Which one is the correct study design?
-
The description of the study participants is very insufficient. The authors should clarify which organization(s) the workers were recruited from, how participants were invited to the study (via online? offline?), how many were initially enrolled, and how many were excluded based on the criteria. While some of this information is shown in the figure, it should also be clearly described in the text.
-
The authors mention that a “90-day PureHealth Longevity Program” was conducted, but there is no description of what this program entailed. Detailed information about the components of the intervention, frequency, mode of delivery, and participant engagement is essential.
-
Please provide a rationale or justification for the selected sample size.
-
The manuscript does not explain how the primary endpoint and secondary outcomes were chosen. What was the rationale for selecting these outcomes? What exactly is the “self-reported health questionnaire”? What specific tests are included in the “blood test”?
-
Grammatical error:
As almost all data was non-normally distributed and outcomes were as- sessed using non-parametric statistics -
The authors state that they used the Kruskal-Wallis test for unpaired data and the Wilcoxon test for paired data. However, this is a repeated-measures study, so it is unclear why unpaired data analysis is needed. The authors should clarify which data were unpaired and which were paired, and explain how differences between baseline and follow-up measurements were analyzed.
In conclusion, the Methods and Introduction sections are too brief. Unfortunately, the authors appear to have devoted minimal attention to providing readers with the necessary information to understand how the study was conducted. Without clear and detailed methodological descriptions, the scientific rigor and credibility of the study are significantly undermined.
Comments on the Quality of English LanguageGrammatical errors should be thoroughly reviewed and corrected throughout the manuscript.
Author Response
Thank you so much for your review and your feedback. Please check our new, revised manuscript (v2), and comments below:
|
Reviewer 2 |
Comments |
Authors’ Comments |
|
1 |
Thank you for the opportunity to review this manuscript. The study investigates the effects of a 90-day workplace-based lifestyle intervention on various cardiometabolic outcomes. This study addresses a topic that is both timely and relevant. However, the manuscript has several critical limitations. |
Thank you for the feedback, we have reviewed the comments of all three reviewers and incorporated the feedback. |
|
2 |
The manuscript lacks an adequate review of prior research and a contextual discussion that would help clarify the rationale and implications of the study. In addition, the Methods section is severely underdeveloped. The authors do not provide sufficient detail regarding how participants were recruited or what the intervention entailed, making it difficult for readers to understand how the study was conducted. |
Further details have been provided in the revised manuscript (v2). |
|
3 |
While several previous studies have examined the effects of workplace-based lifestyle interventions on health, the current manuscript does not sufficiently reference or discuss these findings. The Introduction should be expanded to include a comprehensive overview of relevant prior research and to clearly highlight the research gap this study aims to address. |
The Introduction and Conclusion sections have been revised accordingly. |
|
4 |
The aim or objective of the study is not clearly stated in the final paragraph of the Introduction |
This comments has been incorporated in the new, revised draft manuscript (v2). |
|
5 |
There is a typographical error: "This single-arm study observational study". Which one is the correct study design? |
Thank you for highlighting this, incorporated in the new, revised manuscript (v2). |
|
6 |
The description of the study participants is very insufficient. The authors should clarify which organization(s) the workers were recruited from, how participants were invited to the study (via online? offline?), how many were initially enrolled, and how many were excluded based on the criteria. While some of this information is shown in the figure, it should also be clearly described in the text. |
This information is included in the new, revised manuscript (v2). |
|
7 |
The authors mention that a “90-day PureHealth Longevity Program” was conducted, but there is no description of what this program entailed. Detailed information about the components of the intervention, frequency, mode of delivery, and participant engagement is essential. |
Further information related to the intervention has been included in the new, revised manuscript (v2). |
|
8 |
Please provide a rationale or justification for the selected sample size. |
Amended (see Section 2. Materials and Method). |
|
9 |
The manuscript does not explain how the primary endpoint and secondary outcomes were chosen. What was the rationale for selecting these outcomes? What exactly is the “self-reported health questionnaire”? What specific tests are included in the “blood test”? |
Thank you for your comment. The new, revised manuscript (v2) has dealt with this. |
|
10 |
Grammatical error: As almost all data was non-normally distributed and outcomes were as- sessed using non-parametric statistics |
Amended. |
|
11 |
The authors state that they used the Kruskal-Wallis test for unpaired data and the Wilcoxon test for paired data. However, this is a repeated-measures study, so it is unclear why unpaired data analysis is needed. The authors should clarify which data were unpaired and which were paired, and explain how differences between baseline and follow-up measurements were analyzed. |
Most importantly virtually none of the data are normally distributed. In relatively large sample sizes these tests are more robust and do not reply on defining a finite population mean. We acknowledge that we could have used paired data analysis, but outliers may have had more influence of this outcome. We feel it is statistically more stringent with lower risk of a type I errors to use nonparametric statistics. |
|
12 |
In conclusion, the Methods and Introduction sections are too brief. Unfortunately, the authors appear to have devoted minimal attention to providing readers with the necessary information to understand how the study was conducted. Without clear and detailed methodological descriptions, the scientific rigor and credibility of the study are significantly undermined |
Thank you for the feedback, as you will be able to note, the new, revised manuscript (v2) has incorporated this feedback and the major revisions have improved the scientific credibility. |
Reviewer 3 Report
Comments and Suggestions for Authors
This is a generally well conducted and well presented study.
My major concerns are:
- It is not clear whether or not the study was confined to the Abu Dhabi Health Services Company as was the referenced pilot study. If so how are its employees distributed over the UAE. How many employees refused to participate.
- In the absence of a control group one would like to know the effects of general public health advice and communications from personal physicians and whether or not there was significant possible bias because of this.
- fig 3 does not show differences in blood parameters between days 0 and 90 but it is claimed that the differences by weight group remained at day 90.
- A similar workforce study not subject to these biases is Leigh and Harrison 1991 JOHS ANZ 7:467-472 .

Author Response
Thank you for your comments. Please see our comments below:
|
Reviewer 3 |
|
Authors’ comments |
|
1 |
This is a generally well conducted and well presented study. |
Thank you for your comments, much appreciated. We believe that this is the first of its kind study at this scale in the region. |
|
2 |
It is not clear whether or not the study was confined to the Abu Dhabi Health Services Company as was the referenced pilot study. If so how are its employees distributed over the UAE. How many employees refused to participate. |
This has now been amended with a more detailed description of the study population in Section 2 (Study design and participants). |
|
3 |
In the absence of a control group one would like to know the effects of general public health advice and communications from personal physicians and whether or not there was significant possible bias because of this. |
Noted. For the general public, it may be more challenging to measure the impact of programs such as these and would require a prospective, international study with control arm. |
|
4 |
fig 3 does not show differences in blood parameters between days 0 and 90 but it is claimed that the differences by weight group remained at day 90. |
We have incorporated your comments into the revised Figure 3 |
|
5 |
A similar workforce study not subject to these biases is Leigh and Harrison 1991 JOHS ANZ 7:467-472 . |
Noted, thank you for sharing this study (Reduction of ischaemic heart disease risk factors following direct probabilistic risk communication in the workplace, J Leigh, J Harrison - Journal of Occupational Health & Safety, 1991), we have strengthened the Introduction with additional referenced related to similar, recent studies |
Round 2
Reviewer 2 Report
Comments and Suggestions for Authors
Thank you for addressing my comments. I have some concerns left:
- Although some additional details have been provided, the description remains insufficient. Specifically, what kinds of healthy eating or healthy living promotion interventions were implemented? What stress management strategies were included? What information or interventions did the mobile application deliver? What wearable devices were used? Were these interventions standardized across participants? Further clarification is needed to present a more concrete and comprehensive description of the PHLP and its specific components.
- Lines 297–304: The paragraph suddenly shifts to a discussion of Japan, despite the study being conducted in the UAE. The relevance and rationale for including this section are unclear.
Author Response
|
Reviewer 2 |
Comments |
Authors’ Comments |
|
1 |
Although some additional details have been provided, the description remains insufficient. Specifically, what kinds of healthy eating or healthy living promotion interventions were implemented? What stress management strategies were included? What information or interventions did the mobile application deliver? What wearable devices were used? Were these interventions standardized across participants? Further clarification is needed to present a more concrete and comprehensive description of the PHLP and its specific components. |
Thank you for the feedback. We have addressed these comments in Section 2, Materials and Methods |
|
2 |
Lines 297–304: The paragraph suddenly shifts to a discussion of Japan, despite the study being conducted in the UAE. The relevance and rationale for including this section are unclear. |
The discussion section has been rewritten to enhance the relevance and rationale. |